# The Evolution of Medical Countermeasures for Ebola Virus Disease: Lessons Learned and Next Steps

**DOI:** 10.3390/vaccines10081213

**Published:** 2022-07-29

**Authors:** Ian Crozier, Kyla A. Britson, Daniel N. Wolfe, John D. Klena, Lisa E. Hensley, John S. Lee, Larry A. Wolfraim, Kimberly L. Taylor, Elizabeth S. Higgs, Joel M. Montgomery, Karen A. Martins

**Affiliations:** 1Clinical Monitoring Research Program Directorate, Frederick National Laboratory for Cancer Research, Frederick, MD 21702, USA; ian.crozier@nih.gov; 2U.S. Department of Health and Human Services (DHHS), Assistant Secretary for Preparedness and Response (ASPR), Biomedical Advanced Research and Development Authority (BARDA), Washington, DC 20201, USA; kyla.britson@hhs.gov (K.A.B.); daniel.wolfe2@hhs.gov (D.N.W.); john.lee@hhs.gov (J.S.L.); 3U.S. Department of Health and Human Services (DHHS), Assistant Secretary for Preparedness and Response (ASPR), Biomedical Advanced Research and Development Authority (BARDA), Oak Ridge Institute for Science and Education (ORISE) Postdoctoral Fellow, Oak Ridge, TN 37831, USA; 4Viral Special Pathogens Branch, Division of High Consequence Pathogens and Pathology, Centers for Disease Control and Prevention, Atlanta, GA 30333, USA; irc4@cdc.gov (J.D.K.); ztq9@cdc.gov (J.M.M.); 5Integrated Research Facility, National Institute of Allergy and Infectious Diseases, Fort Detrick, MD 12116, USA; lisa.hensley@nih.gov; 6U.S. Department of Health and Human Services (DHHS), National Institutes of Health (NIH), National Institute of Allergy and Infectious Diseases (NIAID), Rockville, MD 20852, USA; larry.wolfraim@nih.gov (L.A.W.); kimberly.taylor3@nih.gov (K.L.T.); ehiggs@niaid.nih.gov (E.S.H.)

**Keywords:** Ebola virus, filovirus, medical countermeasure, preparedness, diagnostics, vaccines, therapeutics

## Abstract

The Ebola virus disease outbreak that occurred in Western Africa from 2013–2016, and subsequent smaller but increasingly frequent outbreaks of Ebola virus disease in recent years, spurred an unprecedented effort to develop and deploy effective vaccines, therapeutics, and diagnostics. This effort led to the U.S. regulatory approval of a diagnostic test, two vaccines, and two therapeutics for Ebola virus disease indications. Moreover, the establishment of fieldable diagnostic tests improved the speed with which patients can be diagnosed and public health resources mobilized. The United States government has played and continues to play a key role in funding and coordinating these medical countermeasure efforts. Here, we describe the coordinated U.S. government response to develop medical countermeasures for Ebola virus disease and we identify lessons learned that may improve future efforts to develop and deploy effective countermeasures against other filoviruses, such as Sudan virus and Marburg virus.

## 1. Introduction

Since the SARS-CoV-2/COVID-19 pandemic began in late 2019, the world has experienced the dramatic impact that viruses can have on the economy, entertainment, education, and everyday life. However, SARS-CoV-2 is not the only virus with which the world contends, and the lessons learned over decades of response to viral disease outbreaks have enabled the agile global response to the current pandemic. From 2014–2016, the world’s attention focused on Ebola virus (EBOV), a pathogen archetyped in lay print and film as causing among the “deadliest” of viral diseases. Between investments in product development and preclinical and clinical research at National Institute of Allergy and Infectious Diseases (NIAID), part of the National Institutes of Health; outbreak response and health infrastructure support from the Centers for Disease Control and Prevention (CDC); research and logistics support from the Department of Defense; regulatory review and approvals by the U.S. Food and Drug Administration (FDA); and advanced research and development and product procurement by the Biomedical Advanced Research and Development Authority (BARDA), the United States government (USG) supported remarkable progress in combatting this deadly pathogen. Working in partnership with product developers, international organizations, local governments, and front-line workers, USG supported regulatory approval of a diagnostic test, two vaccines, and two EBOV-specific therapeutics. The lessons learned in responding to Ebola virus disease (EVD) outbreaks have informed future response capacity and helped identify gaps in the preparedness and response arsenal for future disease outbreaks. Here, we highlight some of the biomedical advances made against EVD, identify remaining gaps, and parse the lessons learned that have the potential to improve our response to other filovirus diseases, such as those caused by Marburg virus (MARV) and Sudan virus (SUDV). To focus our efforts, we discuss three major topic areas—diagnostics, vaccines, and therapeutics—in the context of the clinical and field operational settings that are inextricably linked to effective countermeasure deployment. In each area, we summarize lessons learned and look toward an optimal future. While biomedical advances are one component of an array of tools to prevent and minimize the impact of EVD outbreaks, we acknowledge that biomedical tools should not be considered in isolation, but in parallel with other public health, communications, modeling, and ecologic/environmental approaches to prevent, predict, and prepare for outbreaks.

## 2. Diagnostics

### 2.1. Overview of the Ebola Virus Field Diagnostic Testing Laboratory

Rapid in-field detection is critical to responding to rare and re-emerging pathogens. When EBOV was first discovered in 1976, laboratory diagnosis could require weeks, and it entailed international shipment of samples to specialized laboratories, which then evaluated those samples using viral isolation and post-mortem histopathologic studies of biopsied tissues [1]. More than four decades later, a suspected EVD patient can be diagnosed within hours in field laboratories via PCR testing enabled by the Cepheid GeneXpert platform, which received an emergency use authorization (EUA) in March of 2015 [2,3]. Post-mortem cadaveric testing can determine death from EVD even more quickly via the OraQuick Ebola Rapid Antigen Test. This test was authorized for emergency use on 31 July 2015 for use with whole blood and fingerstick blood, and then amended in 2016 for use with oral fluids for cadaveric testing. It ultimately received a de novo 510(k) approval in October of 2019. The diagnostic agility provided by these two tests alone reduces the potential for the spread of the disease in the community by enabling early infection prevention and control (IPC) and preventive vaccination interventions. In addition, early diagnosis enables earlier clinical and therapeutic interventions, which can reduce the morbidity and mortality associated with EVD for the individual patient [4]. 

Advancements in diagnostics have facilitated the decentralization of testing. Use of assays that are simple to use and require minimal training and equipment means testing is no longer restricted to a central reference laboratory. Field laboratories have been adapted to incorporate biosafety measures that minimize biohazard risks to laboratory workers, permitting the modern field laboratory to provide integrated diagnostic and clinical laboratory testing that was previously not possible [5]. Notably, these field laboratories also enable near-patient point-of-care clinical laboratories that are critical to effective care, as will be discussed below from the perspective of treatment and clinical management. In this section, we discuss current approaches to laboratory diagnostics during an EVD outbreak. We describe the current tools used for the rapid diagnosis of EVD, and we highlight the importance of next generation sequencing, which facilitates decisions about the use of medical countermeasures (MCMs) and enables the molecular epidemiology that is now a standard component of tracking outbreak transmission dynamics. We also discuss the use of EBOV antigen rapid diagnostic testing (RDTs) for EBOV detection in post-mortem surveillance, and we define development needs to broaden the utility of these tests. We contextualize these tools in the operational challenges that impact the effective deployment of field laboratories. Finally, we identify current gaps and anticipate future needs as national laboratory systems continue to adapt to meet the known and unknown demands in outbreak settings.

### 2.2. EVD Diagnostics: EBOV-Specific Assays and Genome Sequencing

The Cepheid GeneXpert platform is a simple to use, closed, real-time reverse transcription polymerase chain reaction (RT-PCR) system that performs the extraction, reverse transcription, and detection of EBOV in under 100 min. The introduction and use of the Cepheid GeneXpert technology has greatly simplified processing and testing of samples from EVD suspects, and the closed system reduces the risk of contamination. During the 2013–2016 Western African outbreak, across-lab comparison important to outbreak management and research was confounded by the use of multiple different RT-PCR platforms that often utilized different genomic targets [5,6,7]. In contrast, the widespread, standardized, and harmonized deployment of GeneXpert in near-treatment unit field labs—often in areas already familiar with the platform as a tool for tuberculosis diagnostics—has significantly advanced patient care, the ability to do clinical research, and the public health response. Challenges remain, including the maintenance of equipment and supplies, the semi-quantitative nature of the reported cycle threshold results, and rapid, accurate diagnosis of the sick EVD suspect who is ruled out for EBOV infection. However, the platform has been the diagnostic work horse in responses to every EVD outbreak since 2016.

Rapid diagnostic tests (RDTs) have the potential to improve the identification and management of EVD outbreaks in an agile and cost-effective way. First studied towards the end of the 2013–2016 Western African outbreak [8], these tools have been subsequently used in both DRC and Guinea for cadaveric surveillance during the 90-day enhanced surveillance period that follows the declaration of the end of an EVD outbreak [9]. Easy to use EBOV RDTs provide a result in 30 min or less from post-mortem oral swabs or intracardiac puncture samples, and they require minimal biosafety and technical training. However, at the time of this writing, none of the commercially available EBOV RDTs had demonstrated a positive percent agreement above 90% for the detection of low amounts of EBOV in blood, thus making them less than ideal for screening samples from living suspect patients, particularly in the early stages of infection [10,11]. As individuals who have died from EVD tend to have high virus titers, RDTs have been useful for post-mortem screening, an activity that is frequently performed as part of the duties of Safe and Dignified Burial teams and often in conjunction with confirmatory PCR testing of samples. Being able to screen and release bodies that test negative for infection is a valuable tool both from a public health perspective and in terms of community engagement, as it enables bodies to be returned to their families for traditional burial. 

First used widely in the 2013–2016 Western African outbreak, advanced genomic sequencing characterization has become a standard part of EVD outbreak response. Though initially this capability was exclusive to central laboratories, some field laboratories have added the capacity, typically using Oxford Nanopore’s MinION technology. Direct sequencing in the field can provide actionable data to identify and characterize transmission chains and assist contact tracing in near real time [12,13]. Furthermore, sequencing analyses performed on patient samples early in an outbreak can provide critical information as to whether currently available MCMs may be effective based on the species and strain of the virus in circulation; the data can also help monitor for the development of resistance to deployed countermeasures [14]. Inclusion of Next Generation Sequencing (NGS) technologies into the field laboratory, however, is not without its challenges, including necessary increases in the laboratory footprint and the requirement that laboratorians with specialized skills be deployed. Continued development and deployment of these technologies, building in-country infrastructure and systems, and investment in the development of in-country expertise have already paid dividends and are expected to be high-yield in the future, if countries can sustain the resources and retain expertise outside of outbreak response. 

### 2.3. Diagnostics: Operational Aspects

In the last decade, several countries in Eastern (i.e., Uganda), Central (i.e., Angola, the Democratic Republic of the Congo [DRC]) and Western Africa (i.e., Guinea, Liberia, Nigeria, Sierra Leone) have experienced filovirus disease outbreaks [15]. Countries with frequent outbreaks (e.g., DRC, Uganda) have implemented centralized reference laboratory models to manage testing of samples from patients suspected to be infected with high consequence pathogens, including filoviruses. Centralized models provide several benefits: maintaining appropriate waste management practices, laboratory infrastructure, equipment, and testing reagents in local laboratories is challenging logistically and financially. Additionally, routine training, proficiency, and retention of expert staff is more easily maintained in a centralized facility than it is in disparate field sites. Finally, in locations where financial resources are limited, investing in new technologies at a central location is more reasonable than attempting to equip multiple sites. Utilizing a national, centralized laboratory may therefore be the only approach that allows a country to support the multiple diagnostic platforms, including nucleic-acid amplification systems, serological testing platforms, and NGS, which may be necessary to identify the cause of infection in a suspect patient and the origin of an outbreak. Despite these benefits, the effectiveness of a single, centralized reference laboratory is critically dependent on timely sample transfer from sites where suspect cases are occurring and on well-maintained communications, availability of reagents and supplies, and transport capabilities [16]. Loss of sample integrity due to cold chain disruptions, mislabeling of samples, or transportation challenges can lead to false negative results, delaying outbreak response mobilization. The coordination of activities is essential, and it may be best achieved through a governmental mandate that identifies one agency as responsible for all high-consequence pathogen testing [17].

A variation of the central laboratory model especially important in large countries has been to create additional regional referral network laboratories. These are primarily purposed to decrease the turn-around-time from sample collection to sample reporting and thus improve case detection and contact tracing. In the DRC, for example, the central testing laboratory, Institut National de Recherche Biomédicale (INRB) for Viral Hemorrhagic Fever (VHF), is located in Kinshasa, whereas a second INRB-operated regional laboratory has been established in eastern DRC (i.e., Goma) to support more rapid testing of samples in this region of the country. The coordination of activities through a governmental mandate (e.g., National VHF testing strategy) so that one agency (e.g., INRB) is responsible for coordinating all EVD testing across central and regional testing laboratories remains important. In addition to centralized coordination, adequate and sustained funding are also necessary to maintain continuity of operations on an annual basis.

During an active outbreak, central or regional labs support deployment of field labs near the outbreak epicenter. In recent DRC outbreaks, field laboratories have included one or more Cepheid GeneXpert four-chamber instruments (supported by computer hardware and software) and a glove box for safe sample inactivation prior to sample insertion into the closed cartridge for extraction and amplification. These labs have also routinely provided point-of-care clinical laboratory evaluation critical to effective care (e.g., electrolytes, hepatic and renal function, and basic hematology) as well as malaria rapid diagnostic testing. Basic cold chain requirements (2–8 °C refrigeration and −20 °C freezer) are essential components to the field laboratory. 

Common to all these settings is the importance of laboratory information management systems that protect confidentiality but enable rapid action during an outbreak response and, in retrospective analysis after outbreak end, inform the future. Linking laboratory data to epidemiologic, demographic, and clinical context—optimally on a patient-specific level that includes vaccine or therapeutic receipt—remains an ongoing challenge, especially in the absence of established patient tracking systems. Adding to the complexity, different laboratories may be geographically associated with different health facilities at various stages of the patient flow process, complicating the tracking of patient specific information across sites. Addressing and understanding these challenges in managing laboratory information in prior outbreaks should valuably inform progress. 

### 2.4. Diagnostics: From Lessons Learned to the Optimal Future

While the improvement in diagnostic capabilities and laboratory support for EVD has been significant in recent years, there are numerous ways in which those capabilities could continue to be improved. Investment in affordable, mobile, multi-pathogen diagnostic platforms, fieldable assays targeting likely causative agents, or improved pathogen-agnostic NGS technologies will likely inform our understanding of the causes of disease in remote areas and could help identify local threats before they become international threats. It is notable that most patients presenting for care with clinical signs or symptoms suggesting EVD do not actually have EVD; however, with the exception of malaria, rapid diagnosis of other infectious causes in EBOV-negative suspects remains a formidable challenge for clinicians. Providing diagnostic testing for a broader range of pathogens may additionally improve community trust and ultimately enhance early arrival to care/treatment, early isolation and IPC intervention, and early contact tracing. 

In parallel to high-tech advances, the optimal future might include improvements in lower-tech, highly sensitive and specific pathogen-specific RDTs. Development of more sensitive RDTs that could be used to accurately rule in/out disease in the pre-mortem suspect setting would be an important step forward in EBOV field testing, particularly in circumstances in which it might take days to transport a sample from collection to laboratory or where it is difficult to move a suspect case. Such products could provide nearly immediate results that improve time to isolation, diagnosis, and care initiation in Ebola Treatment Units (ETUs); accelerate contact tracing; and improve community engagement. Moreover, routine use of filovirus RDTs for post-mortem surveillance even outside of outbreak response has the potential to identify disease cases that would otherwise go undetected, speeding the declaration of an outbreak and the resources that follow. In all cases, the sensitivity and specificity of RDTs is critical to understand, as the consequences of false negative or false positive results are high for the individual patient and for public health. In addition, the development of RT-PCR tests and RDT’s with a longer shelf-life (from 9–12 months to 2 years) would enable the distribution and maintenance of diagnostic capacity in high-risk countries where they are needed most, reducing stock shortages, and reducing the need to continually purchase tests and pay for the associated shipping costs. There are several promising new or revised RDTs that could soon be evaluated and may offer the possibility of bringing the lab to the patient’s bedside.

While new assays and technologies might be readily identified, the importance of partnership and long-term investments in developing lab-specific and general health systems strengthening needs to be emphasized as crucial to the optimal future. Generally improving expertise in good documentation, information management, and patient tracking would provide multi-pronged benefit. Moreover, investment in these core competencies may enable future collaborations between affected countries and diagnostic developers to support the development and testing of new assays for regulatory clearance. Since certain samples are difficult to ship (e.g., EBOV samples), field testing and the transmission of that data back to interested parties may enable regulatory/administrative bodies such as the U.S. FDA and the European Commission (EC) to approve or certify investigational products for more wide-spread use. 

## 3. Vaccines

### 3.1. Development and Evaluation of EBOV Vaccines

The 2013–2016 Western African EVD outbreak triggered concerted global efforts to develop vaccines that could help stem the outbreak. As of early 2014, two platforms had been assessed in phase 1 clinical trials [18,19], but there were no active clinical stage EBOV vaccine programs. Throughout the latter half of 2014 and 2015, several vaccine candidates entered clinical development, with a smaller subset progressing to phase 2 and 3 clinical trials by early 2015 [20]. Both the Merck product ERVEBO, a recombinant, replication-competent, vesicular stomatitis virus-based vaccine expressing the EBOV (sp. Zaire ebolavirus) glycoprotein (rVSV-ZEBOV), and a Janssen vaccine, a heterologous dose-regimen of Zabdeno (Ad26.ZEBOV-GP) and Mvabea (MVA-BN-filo), emerged and progressed to licensure either through the U.S. FDA or EC using two different regulatory pathways as discussed below. 

ERVEBO was originally developed by the Public Health Agency of Canada, and a partnership with NewLink Genetics helped advance the product before it was licensed to Merck. ERVEBO was evaluated in a phase 3 ring vaccination trial in Guinea during the outbreak in Western Africa [21]. This trial utilized a surveillance-containment strategy to evaluate the effect of vaccination among contacts and contacts-of-contacts of recently confirmed cases. Participants were randomized into two arms—immediate vaccination or delayed vaccination (21 days later)—and the primary outcome was a laboratory confirmed EVD case ten or more days after vaccination [21]. This ten-day threshold was selected based on the anticipated time it would take for an individual to mount an effective immune response to the vaccine. Substantial protection was demonstrated against EVD, with no cases observed among vaccinated individuals from day 10 after vaccination in randomized clusters, as well as in vaccine recipients in non-randomized clusters, albeit with wide confidence intervals (the 95% confidence interval was calculated as 68.9 to 100 percent [21]). The trial demonstrated that it was feasible to evaluate vaccine efficacy in an outbreak setting using ring vaccination (or similar strategies), and it provided vaccine efficacy data that was integral for licensure [21,22]. Ultimately, the World Health Organization (WHO) Strategic Advisory Group of Experts (SAGE) on immunization recommended use of this vaccine in 2017 under an expanded access protocol (EAP) framework, followed by pre-qualification in 2019 and subsequent licensure of ERVEBO by the U.S. FDA in December 2019 [23]. 

The Janssen vaccine, a heterologous dose-regimen of Zabdeno (Ad26.ZEBOV-GP) and Mvabea (MVA-BN-filo), took a different pathway to licensure, in part due to questions of the appropriateness of using a two-dose vaccine regimen in the “reactive” ring-vaccination posture required in an outbreak setting. The vaccine elicited robust antibody responses, which appeared to correlate with protection against disease in animal models [24,25]. That response, in combination with the potential that a two-dose vaccine might elicit increased duration of protection, recommended the two-dose regimen for use in a “preventive” posture in peri-outbreak areas or in health care workers (HCW) in high-risk neighboring countries. Since a phase 3 clinical efficacy trial has not been possible, immunobridging was used to demonstrate clinical benefit, comparing non-human primate (NHP) protection and immunogenicity data to human immunogenicity data [25]. During the 2018–2020 DRC outbreak, phase 3 effectiveness studies of this candidate in peri-outbreak areas were initiated but were adversely impacted by the COVID-19 pandemic; immunogenicity data from vaccinated individuals is likely to inform durability [26]. Ultimately, the Janssen vaccine was granted marketing authorization by the EC in July 2020 and was recommended by the WHO’s SAGE on immunization for use during outbreaks for individuals at some risk of EBOV exposure and preventively, in the absence of an outbreak, for national and international first responders in neighboring areas or countries to which an outbreak might spread. 

The Merck and Janssen EBOV vaccine development programs provided some key insights regarding single vs. two-dose vaccine regimens, onset of immunity or protection, and pathway to licensure. Single dose regimens that confer rapid immunity are preferred in the reactive response to outbreaks; such vaccines may utilize ring vaccination or similar strategies to demonstrate efficacy and are easier to operationalize. In contrast, two-dose vaccine regimens may be more challenging to evaluate during an outbreak, particularly in resource constrained settings or areas experiencing conflict. For two-dose products developed for rare or sporadic indications, the regulatory approach may necessarily follow a “non-traditional” pathway to licensure, such as the European Medicines Agency Exceptional Circumstances or the U.S. FDA Animal Rule [27]. 

### 3.2. Duration of Protection

Duration of protection for the licensed vaccines remains unknown. The rapid onset of protection conferred by ERVEBO enables protection of those at high risk of exposure; however, the durability of protection is a key consideration for health care and front-line workers who may be at risk of exposure for the duration of the outbreak. During the phase 3 ring vaccination trial, the risk of infection declined over time as case numbers improved, preventing the assessment of the durability of protection in vaccinated individuals [21].

Vaccine deployment strategies in recent outbreaks have, to a degree, mirrored the phase 3 clinical trial in efforts to identify index cases followed by the immediate vaccination of their contacts as well as contacts of contacts; continuing to track and characterize breakthrough infections in vaccinated subjects may help inform duration of protection [21]. Vaccination is also recommended for healthcare workers; however, healthcare workers are likely to be revaccinated if a new outbreak occurs, making data from that cohort difficult to interpret [28]. Based on the continued clinical development of ERVEBO [29], antibody responses, measured both by binding and neutralizing antibodies, peak around 28 days post-vaccination; however, measurable titers are still observed in most subjects at least two years after vaccination [30,31,32,33,34]. While this suggests some persistence of immunity, immune correlates for the vaccine are not defined, making interpretation of the data challenging; given the high case fatality ratio of EVD, it may be best to not assume a durable response and re-vaccinate if and when individuals are subjected to a high risk of exposure. 

### 3.3. Post-Exposure Prophylaxis

The initial development of VSV-based filovirus vaccines included nonclinical study evaluation as post-exposure prophylaxis (PEP), demonstrating partial protection when administered soon after relevant filovirus infections [35,36]. Although ERVEBO is not licensed for use as a PEP, nor is it used in subjects recently diagnosed with EVD, the vaccine is used in a reactive “post-event” ring-vaccination setting, and as such is likely administered to many contacts who have already had contact with infectious EBOV. The phase 3 efficacy study indicated that no new cases of EVD appeared in subjects after ten days post-vaccination, indicating a very rapid onset to protection [21], but infections did occur prior to ten days. At this stage, with over 300,000 doses of the vaccine having been utilized in response to outbreaks since 2018, a thorough and updated analysis of EVD cases that have occurred in the first ten days post-vaccination would be critical to inform how rapidly protective immunity develops as well as determine the impact of vaccination on subsequent disease severity and outcomes. 

### 3.4. Immune Assays

Development and standardization of assays to evaluate the immune responses to vaccine candidates was a critical part of the USG response to the EVD outbreak in Guinea, Sierra Leone, and Liberia in 2014–2016. Development and use of common assays and critical reagents facilitated streamlining of efforts and allowed comparison of immune responses across different vaccine platforms and vaccine candidates. This also streamlined regulatory submissions by the various vaccine developers as the US Government filed a Type V Drug Master File with the U.S. FDA that could then be cross-referenced by vaccine developers as part of their submissions to regulatory authorities. The Drug Master File sponsor is the Office of Regulatory Affairs of the US Army Medical Research and Development Command. Assays to evaluate the immune response to vaccination are critically needed as surrogate endpoints that may predict protection and waning efficacy over time. In cases where sufficient clinical efficacy data cannot be collected, appropriate immune assays can serve to bridge protective efficacy and immunogenicity from preclinical NHP challenge studies to human immunogenicity data. This analysis allows scientists to predict likely clinical benefit and support licensure using ‘non-traditional’ regulatory pathways when human efficacy trials are not possible. To serve as a bridge between NHPs and humans, immune assays must be species-neutral, which is demonstrated by establishing parallelism between NHP and human reference standards (RSs) and test samples (TSs). In addition, it is essential to select an appropriate secondary conjugate, which can be shown to fully cross-react with both human and NHP serum. Finally, because the assays would be used as primary immunogenicity endpoints and correlates of protection in phase 3 clinical trials, qualification and validation is required in the labs running the assays. 

The approach taken within the context of EVD was to develop, standardize, qualify, and validate a single immune assay—a human anti-EBOV glycoprotein (GP) IgG Enzyme-Linked Immunosorbent assay (ELISA) that used the recombinant GP from *Zaire ebolavirus* (Kikwit) as the coating antigen [37]. Several generations of RSs were used to support assay development, all of which were derived from immune sera from subjects who had received an EBOV vaccine either as post-exposure prophylaxis or in clinical trials. As additional EBOV vaccine candidates entered clinical trials, larger quantities of vaccinee sera became available. Immune sera were pooled from 371 volunteers participating in clinical trials sponsored by four vaccine developers, where individuals were vaccinated with VSVΔG-ZEBOV (BPSC1001), Ad26.ZEBOV+MVA-BN-Filo, and EBOV GP Nanoparticle adjuvanted with Matrix-M^TM^. Quality controls were also generated in parallel to the reference standard using serum from subjects vaccinated with VSVΔG-ZEBOV (BPSC1001) and rVSVΔG-ZEBOV-GP (V920). The ELISA was ultimately validated [38]. 

The large number of samples collected to support multiple vaccine development efforts necessitated technology transferring the validated ELISA to multiple sites, some of which were in western Africa. Critical to successful technology transfer was the use of common proficiency panels of immune sera that were tested in a blinded fashion by the receiving laboratories; the use of the same standard operating procedure; shared critical reagents and data analysis methods; and frequent and close communication between laboratories. Performance of the human anti-EBOV GP IgG ELISA was tracked across five different laboratories using the proficiency panel, the results of which were published in an interlaboratory study [37]. The results from this study confirmed that results were similar when using the assay at multiple labs. The validated human anti-EBOV GP IgG ELISA was shown to be suitable for testing immune sera from NHPs immunized with EBOV vaccine candidates as well through a formal demonstration of parallelism between the RSs and NHP TSs [39]. The assay was therefore suitable for bridging immunogenicity datasets from NHPs to humans. 

The same ELISA platform used for the anti-EBOV GP IgG ELISA was applied to the development and qualification of anti-MARV and anti-SUDV GP IgG ELISAs. A full description of these assays can be found in the publication by Rudge et al., in this Special Issue. 

### 3.5. Vaccines: From Lessons Learned to the Optimal Future

The EBOV vaccine experience yielded two safe and effective vaccines, but many questions remain. For the Janssen vaccine, better understanding of the durability of protection may help guide use of this product for those at highest risk of multiple viral exposures, including healthcare workers. For ERVEBO, while duration of protection remains an important question, information about the onset of protection and correlates of protection may be more pressing. Significant information on these questions could be obtained from preclinical studies. Investment in animal models should be made to define correlates of protection and assess onset and duration of protection. 

ERVEBO vaccine effectiveness was confirmed through clinical trials, but signals of vaccine breakthrough after ten days have been reported, albeit often in single observations or self-reported by patients [4,40]. A concerted effort to identify and characterize breakthrough infection and disease is crucial to understand true vaccine effectiveness given the individual and public health consequences of even rare events. Turning specifically to PEP, characterization of EVD cases that occur within the first ten days after vaccination could help determine whether vaccination confers an impact on disease severity or outcomes in patients. This information could guide decision-making regarding the use of vaccine vs. therapeutics as PEP in high-risk contacts, which is currently an active area of discussion in the research community and field. For a rare, high consequence disease like EVD, optimizing use of vaccination registries to maximize our ability to learn from existing data is critical, and it would help researchers understand the real-world effectiveness of the vaccines. 

There are a number of lessons learned from our experience developing and ultimately validating the anti-EBOV GP IgG ELISA. Absent a large filovirus disease outbreak caused by SUDV or MARV, quantities of human immune sera to support immune assay development will be limited. First-generation MARV and SUDV ELISA RSs consisted of convalescent sera collected from a small number of survivors from previous outbreaks. As second-generation RSs and prior to initiation of larger clinical trials, we have used purified IgG generated from human transchromosomal cows (Tc bovine) as a bridge to support qualification of the MARV and SUDV assays [38,41,42,43]. Now that MARV and SUDV vaccine candidates have entered clinical trials, larger quantities of vaccine serum can support the creation of a third-generation RS. Pooling vaccine sera from multiple clinical trials to create a single RS is an efficient way to support the programs of multiple vaccine developers. These actions to prepare to support MARV and SUDV assay—and therefore vaccine—development were made possible from the experience of the EBOV vaccine development process and are a testament to the immense value that can be obtained by USG interagency and private sector collaboration.

## 4. Therapeutics

### 4.1. Therapeutics: The First Four Decades (1976–2016)

In 2006, when the Department of Homeland Security determined EBOV was a material threat to national security, there were no approved therapeutics for EVD and treatment was limited to supportive care. Indeed, the therapeutic landscape changed very little in the three decades following the initial characterization of the virus and disease in 1976 [1]. However, this terrain has evolved greatly in the last two decades. Under emergency use authorization, EBOV-specific therapeutics were first administered to patients in the U.S. and Europe beginning in 2014, and the unprecedented 2013–2016 Western African outbreak also provided an opportunity to test investigational therapeutics in clinical studies. Eight non-randomized single-arm intervention studies, unfortunately uncontrolled or often using historical controls, investigated the efficacy of convalescent whole blood or plasma, interferon β-1a, favipiravir, the antimalarial artesunate-amodiaquine, brincidofovir, and TKM-130803 [44]. In addition, the first RCT of an EVD therapeutic was initiated in 2015 by the Partnership for Research on Ebola Virus in Liberia (PREVAIL) working group [45]. This study was designed to evaluate the efficacy of Mapp Biopharmaceutical’s ZMapp, a cocktail of three monoclonal antibodies (mAbs), for the treatment of patients with a PCR-confirmed diagnosis of EVD; however, the number of patients enrolled was limited by the rapid decline in eligible new cases of EVD and a determination of product efficacy could not be made [45]. Despite inadequate enrollment, only 22% of patients who received ZMapp plus standard of care succumbed to disease as compared to 37% of those receiving standard of care alone, and this trend toward efficacy enabled consideration of ZMapp as a standard of care arm in future clinical studies [4]. Lessons learned from this research experience, highlighted by a National Academy of Sciences after-action report [46], included agreement that randomized controlled trials were ethical, appropriate, and the most efficient way to provide the best possible information on efficacy and safety. Consistent with this recommendation, meta-analytic efforts to compare patient-level data across therapeutic studies from Western Africa were confounded by the absence of adequate control groups and randomization [47]. There was a clear challenge to the research community implied in the National Academy of Sciences recommendation to enable clinical research agility and initiate well-designed, optimally pre-positioned studies during active outbreaks [46].

### 4.2. Therapeutics: The PALM RCT (2018–2020)

Between 2018–2020, the second largest EVD outbreak in recorded history occurred in the provinces of North-Kivu and Ituri in the DRC [48]. Initially, four EBOV-specific therapeutics were made available to patients under a WHO/DRC monitored emergency use of unregistered and experimental interventions (MEURI) EAP that was intended to bridge to a clinical trial. Building on the Western African experience, a NIAID/DRC INRB-led multi-partner collaboration (DRC Ministry of Health, WHO, non-governmental organization (NGO) partners, and pharmaceutical companies) rapidly developed a clinical trial protocol such that the Pamoja Tulinde Maisha (PALM [“Together Save Lives” in the Kiswahili language]) RCT was initiated only four months after the outbreak was declared. Patients were randomly assigned in a 1:1:1:1 ratio to one of four treatment arms: (i) ZMapp, considered the control group due to the results of the PREVAIL II study [45]; (ii) remdesivir, a nucleotide analogue RNA polymerase inhibitor; (iii) REGN-EB3, a three mAb cocktail; and (iv) mAb114, a single mAb [4]. On 9 August 2019, an interim analysis based on 681 patients showed that the REGN-EB3 arm had crossed an interim boundary for efficacy and both REGN-EB3 and mAb114 appeared to be superior to treatment with ZMapp or remdesivir; in response to the data, the safety monitoring board recommended that all additional patients be assigned to the mAb114 and REGN-EB3 groups in an extension phase of the trial. 

Partially on the basis of the PALM RCT results, the U.S. FDA approved Biologics License Applications for REGN-EB3 (trade name: Inmazeb) on 14 October 2020, and mAb114 (trade name: Ebanga) on 21 December 2020, for the treatment of patients with EVD, making these the first and only licensed therapeutics for a filovirus disease indication [49,50]. Inmazeb is a cocktail of three mAbs (REGN3470, REGN3471, and REGN3479) developed via immunization of VelocImmune^®^ mice. These antibodies bind to three unique epitopes on the EBOV glycoprotein (GP) resulting in viral neutralization and activation of antibody dependent cellular cytotoxicity functions [51]. Ebanga is a single mAb isolated from the blood of an EVD survivor, which targets the receptor-binding domain of the EBOV GP, preventing endosomal release of the viral genome into host cells [51]. Considering that Inmazeb and Ebanga target distinct epitopes, the two products provide insurance against the development or emergence of a virus strain that is resistant to both products.

### 4.3. After the PALM RCT: Operational Challenges for Therapeutic Delivery

In 2020, an EVD outbreak in DRC’s Equateur province overlapped with the tail end of the North-Kivu/Ituri outbreak; this outbreak occurred immediately after the PALM trial results were made known, mobilizing the community to ensure that Ebanga and Inmazeb were made available to patients. However, the geographic breadth of this outbreak made it challenging to quickly deliver effective care and therapeutics to patients. Overall, of 130 confirmed and probable EVD cases, only 78 patients actually arrived at treatment, transit, or isolation centers; of those, 32 patients received Inmazeb or Ebanga. Though observational (these were not randomized comparisons), the case fatality ratio (CFR) was much improved in patients receiving EBOV-specific treatment (2/32 = 6% CFR) compared to patients untreated (53/98 = 54% CFR) [52]. Shortly after their U.S. FDA approval, Inmazeb and Ebanga were deployed to combat several EVD outbreaks in 2021 in Guinea and DRC (Table 1). In these outbreaks, observational data suggested that the use of therapeutics reduced the case fatality rates and, between the rapid public health response and use of ERVEBO in ring vaccination campaigns, end of transmission was achieved within 105 days, on average. Importantly adding to the 2018–2020 experience, these 2021 outbreaks demonstrate a commitment to a new standard of care that routinely involves these approved therapeutics.

The execution of the PALM study and approval of Inmazeb and Ebanga changed the anticipated regulatory strategy for filovirus MCM development. While the path to licensure for filovirus therapeutics in the U.S. had been anticipated to be through the U.S. FDA’s Animal Rule [27], the PALM RCT demonstrated that an outbreak clinical efficacy trial was feasible, albeit challenging. Approval of Inmazeb and Ebanga and their now standard use in EVD case management has shifted the dynamic of EVD outbreaks; however, despite these advances, significant gaps still exist in the therapeutic space.

### 4.4. After the PALM RCT: Improving Outcomes in Severe Disease

In the PALM RCT, death (up to 28 days) occurred in 33.5% of the Inmazeb group (*p* = 0.002) and in 35.1% in the Ebanga group (*p* = 0.007) as compared with 49.7% overall in the ZMapp control group [4]. CFR in the treatment groups was higher in those presenting with higher viral loads and more severe disease: 67% of patients who presented with higher viral loads (as proxied by Ct value ≤ 22.0) died despite receiving either Inmazeb or Ebanga [4]. These data likely reflect the importance of early detection, diagnosis, and admission to enable rapid treatment, as well as the need to improve supportive care. However, poor outcomes in these patients also query the potential for further improvements to the EBOV-specific therapeutic armamentarium. Certainly, the PALM RCT data suggests that early delivery of these mAb-based therapeutics increases their effectiveness; the addition of a second EBOV-specific therapy to Inmazeb or Ebanga treatment could plausibly improve outcomes. Possible approaches include combining two effective mAb-based therapeutics or considering optimization of potency, dose, and/or Fc effector function of current products. However, outcomes might be more likely improved by combining effective mAb products with a mechanistically independent second product, potentially a small molecule direct-acting antiviral able to inhibit viral replication in tissues of interest, especially in the setting of widespread dissemination of EBOV. 

The evaluation of any additive/synergistic effects for a combination approach for EVD is challenging. At a minimum, the demonstration of non-interference between proposed products and the absence of any major safety signal should be evaluated. Efficacy may be more difficult to establish, however. The filovirus NHP models were developed to be universally or nearly universally lethal; while this stringent high bar provides a clear efficacy signal in NHPs, the very narrow treatment window makes evaluation of combination therapeutics challenging. The added benefit of a second product would likely need to be demonstrated by extending the therapeutic intervention window past the current day five treatment benchmark. A recent study in the NHP model of MARV disease provided proof-of-principle that combination therapy with mAbs and remdesivir may provide a survival benefit [53]. NHPs challenged with a lethal infection of MARV followed by treatment 6 days post-inoculation (dpi) with MR186-YTE (a mAb targeting the MARV GP) or remdesivir alone resulted in 0% survival (remdesivir *n* = 0/5; MR186-YTE *n* = 0/5). However, treatment with both products in combination beginning 6 dpi showed 80% survival (*n* = 4/5). A similar result was seen when remdesivir was used in combination with a SUDV mAb cocktail MBP431 [54]. These data support further testing of combination therapies toward decreasing the case fatality of acute filovirus disease and urge similar evaluation in NHP EBOV challenge models. Finally, careful deliberation around the optimal current or future candidates and trial design for evaluation of combination approaches in a future outbreak clinical trial setting is ongoing.

Efforts in the COVID-19 pandemic toward improving patient outcomes have been instructive in elucidating key roles for both virus-specific therapeutics and those targeting the host response. The relative contributions of viral and host immunopathologic mechanisms to organ dysfunction and severe disease and death in EVD are yet to be fully understood; nonetheless, our current understanding of EVD pathogenesis suggests that pathogen-agnostic strategies (e.g., immunomodulatory targeting of dysregulated inflammation or host endothelial stabilization) are potential additions to approved virus-targeted therapies to improve patient outcomes. Thus far, biomarking inflammatory and immunopathologic correlates of outcome with higher resolution systematic approaches have been limited to small numbers of human subjects or in animal models [55,56,57,58,59]. However, these limited data recommend more exploration of host-targeted approaches to further improve outcomes. Further study in the pre-clinical animal models and a higher-resolution understanding of human disease (e.g., in the large datasets accompanying the PALM RCT or in those patients treated under the MEURI protocl) could valuably inform this question.

### 4.5. Therapeutics in Context: Optimal Supportive Care at the Clinical Bedside

Effective supportive and critical care remains crucial to improve outcomes in EVD patients, particularly in those presenting with high viral loads and late into severe disease with multi-organ dysfunction/damage syndromes. While early intervention with effective EBOV-specific therapeutics is critical, therapeutic strategies cannot be seen as “magic bullets” in isolation and must be accompanied by the will and capacity to deliver needed clinical support [60]. When compared to prior experience, the 2018–2020 DRC North Kivu outbreak saw a significant upgrade of the ability to monitor and to provide specific and bundled supportive care to patients in novel non-traditional care spaces. Components of this upgrade included innovative care structures designed to enhance the delivery of safe and effective monitoring and care and improve communications between providers, patients, and families; the provision of necessary supplies to enable clinical and laboratory monitoring and supportive care; increased attention to staff training, including the WHO’s rapid in-outbreak development and publication of supportive care guidelines [61]; the development of standard care systems, including “bundled” supportive care; and mechanisms to enhance HCW safety. Crucial to this upgrade was the standard availability of near-patient (or near-treatment center) point-of-care clinical laboratory testing for serial evaluation of electrolytes, renal function, liver injury, biomarkers of inflammation, and hematologic indicators. Additionally, now considered standard is near-patient rapid testing for common coinfections, e.g., Human Immunodeficiency Virus (HIV), malaria RDT, etc.; further research is needed to define the burden of co-infections or incident secondary infections during the course of EVD and their impact on patient outcomes. The outcomes from a few patients with severe disease treated in well-resourced settings during the 2013–2016 Western African outbreak signaled that extra-corporeal organ support could be safely and effectively delivered [62]. The optimal future requires further understanding of the clinical phenotypes (especially the determinants of organ dysfunction) associated with EVD as well as advances to prevent and support that organ dysfunction in a field setting, most notably in the setting of acute kidney injury. 

### 4.6. Therapeutics: Viral Persistence

One of the earliest descriptions of persistent infection in a filovirus disease survivor originated in the 1969 publication summarizing the first recorded outbreak of Marburg virus disease (MVD) [63,64]. In this outbreak, MARV persisted in the seminal fluid of a convalescent patient, resulting in the infection of their partner approximately 60 days after their recovery [63]. Subsequently, preclinical data in macaques and ferrets have demonstrated that persistent infection can be observed across animal models of infection. While this phenomenon was known, the unprecedented number of EVD survivors of the Western Africa outbreak enabled characterization of EBOV persistence in larger cohorts and included well-documented reports confirming the implications for public health (sexual transmission and reignition of outbreaks) and individual survivors (uveitis and meningoencephalitis associated with EVOV persistence) [65,66,67,68,69]. Interest has continued in subsequent outbreaks: WHO and the DRC INRB, among others, have launched public health efforts tracking survivors of the DRC 2018–2020 EVD outbreak, including for clinical sequelae and for viral RNA persistence in reproductive fluids. At the tail end of the 2018–2020 DRC outbreak, the report of a fatal “relapsed” EVD case in an EVD survivor six months after treatment with mAb114 (i.e., Ebanga) highlighted the unfortunate collision of these individual and public health concerns; this case led to more than 90 subsequent human-to-human transmissions over a broad geographic area and extended the outbreak and response efforts an additional six months [40].

In 2021, though largely overshadowed by the SARS-CoV-2/COVID-19 pandemic, three separate EVD outbreaks occurred in DRC or Guinea, in addition to a fourth outbreak of MVD. Remarkably, viral genomic sequencing suggested that the origin of all three EVD outbreaks was likely related to viral persistence in an EVD survivor (e.g., sexual transmission or recrudescence) and not a new spillover event from an animal reservoir [48,67]. The DRC outbreaks were both linked to cases from the 2018–2020 outbreak in North-Kivu and Ituri provinces, and sequencing suggested that the outbreak in Guinea was linked to the 2014–2016 Western Africa epidemic. In particular, the Guinea 2021 lineage shows considerably lower divergence than would be expected during sustained human-to-human transmission, suggesting persistent infection with reduced replication or a period of latency [21]. Previously, EBOV RNA has been detected in semen up to 965 days after acute disease onset, albeit at very low levels; of more importance, sexual transmission related to EBOV in semen had been documented up to 482 days after acute disease onset. These new data from Guinea suggesting viral persistence beyond five years have triggered a paradigm shift in our understanding of EBOV persistence and emphasize the importance of learning from past EVD outbreaks, improving responses in future outbreaks, and supporting EVD survivors. Key questions remain to be answered about the host-pathogen determinants of viral persistence (and associated transmission or recrudescent disease) at the epidemiologic, individual, organ/tissue, cellular, and molecular levels. Furthermore, the interaction of now standard EBOV-specific therapeutics in determining, preventing, or treating viral persistence remain unknown. Limited investigation in NHP models suggests that receipt of particular therapeutics, or classes of therapeutics, may impact viral persistence and associated recrudescent clinical syndromes [70,71]. Further investigation of survivors of the 2018–2020 DRC outbreak, most of whom received an EBOV-specific therapeutic under the MEURI EAP or as part of the PALM RCT, may inform this uncertainty.

With regard to the use of EBOV-specific therapeutics in the setting of viral persistence, data are sparse. EBOV-specific therapeutics have been used under EAP in the previously described patients with EBOV persistence associated with uveitis (favipiravir) and meningoencephalitis (remdesivir), both in conjunction with corticosteroids [68,69]. Regarding clinical studies, NIAID and partners have executed intervention studies aimed at treating persistent infection; a small signal of potential benefit for the use of remdesivir to clear the semen of male EVD survivors needs to be confirmed in larger studies [72]. 

### 4.7. Therapeutics: Post-Exposure Prophylaxis

Current WHO guidance for PEP varies based on assessment of risk, but it focuses on “ring vaccination” with the U.S. FDA approved rVSVΔG-ZEBOV-GP vaccine (trade name ERVEBO), in part due to the rapid onset of protection conferred by the vaccine. However, as a vaccine, ERVEBO is anticipated to provide protection only after a minimum of ten days, at which point a vaccine-elicited immune response can be detected [21]. Due to the rapid progression of filovirus disease, ten days is an unacceptable window for someone who has been exposed to the virus and is infected. A PEP product in the EVD armamentarium would be expected to have numerous benefits, including reduced mortality, faster interruption of transmission chains, greater willingness of close contacts to present themselves, and greater likelihood of controlling outbreaks at their source. 

Historically, though absent research evidence, expert consensus has informed the use of EBOV-specific therapeutics after health-care-worker or laboratory high-risk exposures. Advancing the use of the newly approved monoclonal antibody therapeutics for PEP indications, including but not limited to HCW, in contacts with high-risk exposures to Ebola virus is of interest. Observational experience with this use in high-risk HCW exposure settings in DRC has been recently described [73]. One pragmatic consideration for the use of Inmazeb or Ebanga as PEP is that the current supply of both products is limited and therefore large-scale use as PEP would be impossible at this time. In addition, both products require an intravenous (IV) infusion and cold chain requirements make operationalization difficult. From a public health perspective, co-administration of the mAb products and vaccine might negatively impact the effectiveness of both products, plausibly implying that close contacts could not be vaccinated until the mAb product was cleared. Non-interference studies would be needed to determine the presence and duration of any negative interaction and to inform determination of the appropriate staggering between mAbs and vaccination targeting the EBOV-GP axis. 

It is imperative that the research community work to assess alternate therapeutics for use as PEP. An ideal candidate would be available through oral administration and would not interfere with the immune response to vaccination, such that the PEP product and vaccination could be co-administered. Candidate products should be evaluated in a scientifically rigorous way, ideally through randomized controlled trials using vaccination with ERVEBO as a control arm.

### 4.8. Therapeutics: Operational Product Profile

Filovirus disease outbreaks often occur in remote locations in resource constrained settings. Existing infrastructure is typically not sufficient to respond immediately to patients requiring isolation and aggressive interventions, and cold chain access may be unavailable in the early days of response. Experience gained by responding to outbreaks in remote regions has highlighted important product characteristics that would improve drug deployment and effective administration. Products that require single administration (e.g., Inmazeb and Ebanga) are preferable over multiple dose/day administrations (e.g., ZMapp, remdesivir). Single administration is operationally easier for healthcare providers and also reduces the risk of a patient leaving the treatment unit before the treatment course is completed. While both Inmazeb and Ebanga are administered IV, products with oral, intramuscular (IM), or subcutaneous (SC) administration would be a significant improvement over IV delivery, provided they could offer comparable efficacy. IV administration can be challenging in severely ill filovirus disease patients and requires a higher level of expertise in the healthcare provider. Additionally, orally available or even IM/SC products might enable drugs to be provided to sick individuals who refuse admission to a treatment unit; caution is needed, however, as the typical EVD patient requires supportive care only able to be delivered in a treatment unit setting and releasing infected patients to the community increases the likelihood of subsequent virus spread. Finally, establishing cold chain capabilities—often through the provision of mobile freezers from external partners—causes a delay in access to therapeutics. Considering the remote regions in which filovirus disease outbreaks often occur, products that do not require cold-chain transportation would significantly improve operational logistics during an outbreak. Improvements in the infrastructure of health care facilities, particularly in the form of consistent access to electricity or generators, would facilitate storage of therapeutics at the outbreak location and reduce the need to establish transport chains from larger, better resourced areas.

### 4.9. Therapeutics: From Lessons Learned to the Optimal Future 

After the decades of concerted preclinical effort that enabled MCM development, the identification and subsequent regulatory approval of two EBOV-specific mAb therapeutics tested in the PALM RCT was a landmark achievement, notable in its demonstration that well-designed RCTs can be effectively implemented even in challenging outbreak environments. Indeed, after many decades in which the clinical bedside had not remarkably evolved, the now standard delivery of Inmazeb or Ebanga, coupled with significant advances in the supportive care provided in newly designed and capacitated treatment units, represents a paradigm shift in improving overall outcomes in EVD. However, the same data urges continued progress to improve poor outcomes in patients with high viral loads and severe disease. The optimal therapeutic future for acute EVD includes preclinical development and clinical strategies to improve the current treatments, including in combination approaches, in continued development of novel therapeutics with broader product profiles, and in exploration of host-directed (and pathogen-agnostic) approaches. What cannot be overemphasized is that EBOV-specific therapy must not be uncoupled from requisite supportive care; ensuring the availability of and continued enhancement of that care is an obvious priority. 

Sobering recent signals of the individual and public health consequences of EBOV persistence in EVD survivors require investigation of the underlying pathophysiology and evaluating the ability of EBOV-specific therapeutics to prevent, mitigate, or treat those consequences. Indeed, the optimal therapeutic future will require preclinical and clinical research attention on both acute and convalescent infection fronts. Therapeutics with an ability to reduce long term sequelae associated with EBOV infection would improve quality of life in survivors and may also prevent viral persistence associated with sexual or other modes of transmission and reignition of outbreaks. In parallel, determining an operationally and scientifically sound approach to PEP using either current or newly identified countermeasures may slow outbreak spread.

While the PALM RCT demonstrated that the execution of clinical trials in outbreak settings is feasible, the study also established an incredibly high standard that may be challenging to meet in other filovirus disease outbreaks. Operationalizing effective clinical research requires rapid development of agreed-upon research protocols that can be moved appropriately through regulatory approval, trained staff, cold chain monitoring, and high-quality data collection and documentation, among other things. Optimally, the staff, supplies, and system competencies required would be pre-positioned; without support to improve infrastructure and basic research capabilities in high-risk outbreak areas, executing future studies will be extremely challenging. As has been discussed in prior sections, developing countermeasures for pathogens endemic to developing regions without concurrently supporting public health systems reduces the functional utility of those MCMs.

## 5. Conclusions and Future Efforts

In response to the 2013–2016 Western African and subsequent EVD outbreaks, the biomedical research community has enabled impressive development of MCMs to combat this threat. Decades of preclinical work led to clinical research that enabled the regulatory approval and widespread deployment of new diagnostic, therapeutic, and vaccine tools, which allow us to detect, prevent, and counter EVD outbreaks. The Cepheid GeneXpert platform and OraQuick Ebola Rapid Antigen Test have reduced the time required to diagnose EVD patients or perform post-mortem screening from weeks to hours. Importantly, these tests have decentralized testing and supported the growth of field laboratories, which are crucial for effective treatment and clinical management. Genomic sequencing characterization has also become a foundation of the outbreak response as it provides critical information about the species and strain of the virus in circulation, which informs use of MCMs in the field. 

Detection is always the first step of an outbreak response, and vaccine deployment can quickly follow. The licensure of Merck’s ERVEBO vaccine and Janssen’s heterologous EBOV vaccine using Zabdeno and Mvabea, provided critical tools in the fight against EBOV outbreaks. These vaccine development programs provided some key insights into the benefit of single dose regimens due to their comparative ease of use in ring vaccination campaigns and the ability to use more “traditional” regulatory approaches for licensure. Although the Merck and Janssen vaccines are highly protective against EVD, the duration of protection is unknown and individuals can become infected before vaccination campaigns are initiated or prior to onset of immunity, which is why therapeutic development is also essential. The approval of the monoclonal antibody therapeutics Inmazeb and Ebanga filled a crucial need in our outbreak response and our ability to treat EVD patients. Yet, several important gaps remain, such as MCMs to treat or prevent viral persistence and MCMs to utilize as PEP. 

The progress made in EBOV MCMs has required remarkable investment—both financially and in expertise—at all stages of development, but this is merely a down payment for a better future. Investments must continue to be made in characterizing, optimizing, and procuring established MCMs, and next generation products with improved operational and technical characteristics should be advanced both for EBOV and other filoviruses of interest. When possible, vaccines and therapeutics effective for multiple filoviruses should be prioritized to mitigate the risk of novel strains or species. As has been described throughout this manuscript, there are many persistent gaps in our capabilities against EBOV; there are also many lessons learned from the EBOV experience that can be used to improve our approach to related filoviruses (e.g., SUDV and MARV), for which there are no approved MCMs (Table 2). 

Technical advancements and MCMs are critical, but they are impotent if they cannot be utilized effectively. Leveraging the experiences from EBOV, plans should be in place to detect, contain, and treat affected individuals in at-risk countries. Ideally each country would have “live” preparedness plans inclusive of communication plans to access MCMs and detailed Standard Operating Plans on their utilization, as well as trained personal, logistical plans for deployment, established cold chains, tracking systems for MCM expiration and replenishment of critical consumables, emergency funding to be immediately activated to enable response, and ongoing engagement of the at-risk communities. Developing, licensing, and deploying MCMs are important steps to stopping outbreaks where they occur and minimizing lives lost; however, advancements in technology are only valuable in combination with trained personnel, strong communication networks, and effective community engagement. 

## Figures and Tables

**Table 1 vaccines-10-01213-t001:** Summary of recent *Zaire ebolavirus* (EBOV) outbreaks and therapeutic use [Case fatality rate (CFR)].

DRC—11th EVD Outbreak 1 June 2020 to 18 November 2020				
**Confirmed Cases**	**Probable Cases**	**Total Cases**	**Deaths (CFR)**	**Recoveries**
**119**	11	130	55 (42.3%)	75
**Therapeutic Use**	Inmazeb	15	2 (13.3%)	13
	Ebanga	17	0 (0%)	17
**DRC—12th EVD outbreak 7 February 2021 to 3 May 2021**				
**Confirmed Cases**	**Probable Cases**	**Total Cases**	**Deaths (CFR)**	**Recoveries**
**11**	1	12	6 (50%)	6
**Therapeutic Use**	Inmazeb	3	0 (0%)	3
	Ebanga	5	2 (40%)	3
**Guinea—2nd EVD outbreak** **14 February 2021 to 19 June 2021**				
**Confirmed Cases**	**Probable Cases**	**Total Cases**	**Deaths (CFR)**	**Recoveries**
**16**	7	23 *	12 (52.2%)	10
**Therapeutic Use**	Inmazeb	8	0 (0%)	8
**DRC—13th EVD outbreak** **8 October 2021 to 16 December 2021**				
**Confirmed Cases**	**Probable Cases**	**Total Cases**	**Deaths (CFR)**	**Recoveries**
**8**	3	11	6 (55%)	2
**Therapeutic Use ^1^**	Inmazeb	2	1 (50%)	1
	Ebanga	2	1 (50%)	1
**DRC—14th EVD outbreak** **23 April 2022 to 4 July 2022**				
**Confirmed Cases**	**Probable Cases**	**Total Cases**	**Deaths (CFR)**	**Recoveries**
**4**	1	5	5(100%)	0
**Therapeutic Use**	Inmazeb	1	1 (100%)	0

* One patient lost to follow up. ^1^ One high risk contact received Inmazeb as Post Exposure Prophylaxis (PEP) during this outbreak.

**Table 2 vaccines-10-01213-t002:** Lessons Learned from the USG response to EBOV outbreaks.

Area	Lessons Learned
**Diagnostics**	RDTs with high sensitivity and specificity are needed to improve outbreak detection and response Field laboratories capable of supporting diagnostics and clinical laboratory testing for patient care improve outbreak operations and patient outcome Development and implementation of data management tools to accurately collect and store clinical, epidemiologic and laboratory data would ensure valuable data are leveraged for future lessons learned Successful utilization of diagnostics relies on the successful collection and transport of samples, proper storage of the samples using cold chain components, stable supply chain, and rapid transmission of data. Without support for these activities, turn-around times for testing and reporting suffer, which adversely impact community trust
**Vaccines**	Rapid deployment of investigative vaccines during an outbreak is essential to collect real world clinical efficacy data that be used to support licensure. Failure to do so means a reliance on bridging data from NHPs—an increasingly scarce resource—and lengthy timelines to approval (Animal Rule or similar pathway to licensure) Single dose vaccine candidates that elicit rapid onset of protection are easier to operationalize as part of a ring vaccination strategy. The role of multiple dose regimens in proactively preventing infection in highest risk populations or in peri-outbreak settings needs to be determined Enabling the establishment and use of clinical databases to track outcomes of vaccinated subjects is essential to understanding duration of protection and the impact of vaccination on disease severity in break through cases Development of immune assays will likely need multiple generations of reference standards, which requires pre-planning and coordination
**Therapeutics**	Monotherapy may not be adequate to effectively treat severe filovirus infection; combination therapeutic approaches, improved supportive care, and host-targeted therapeutic approaches may be required The prevention and treatment of persistent EBOV infection in EVD survivors requires urgent research attention to improve quality of life of survivors and to prevent outbreak prolongation or reignition. Further preclinical and clinical evaluation of therapeutics for this indication is warranted. Products operationally and scientifically appropriate for PEP in high-risk contacts should be evaluated in preclinical and clinical research Identification and mitigation of potential logistical challenges that will impact deployment and evaluation of MCMs must occur during the advanced development of potential products

## Data Availability

Not applicable.

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
