# Peer review of "The Evolution of Medical Countermeasures for Ebola Virus Disease: Lessons Learned and Next Steps"

_vaccines, 2022, doi:10.3390/vaccines10081213_

Round 1

Reviewer 1 Report

Majors:

Ebola countermeasures are not only restricted to therapeutic/preventive agents and diagnostic tools. The vision should be broader, including the following items, even in its introduction section, before going to what already the world has against EVD (Diagnostics, therapeutic, and vaccines), what about the a. public health (including local and international community’s preparedness, are we improved the risk alarming and communications, multimedia and outbreak about EVD, b. EVD modeling and expand prediction modeling, c. and more important factor the environmental and/or ecological effective factor.

Minors:

1.      Table 1 and 2 presentations should be improved.

2.      Hope you add a conclusion section for the current situation before your impact on future efforts section.

3.      Carefully revise the typos

Author Response

Please see our response in the attached PDF. 

Reviewer 2 Report

The authors discussed three major topic areas - diagnostics, vaccines, and therapeutics – which are inextricably linked and crucial to effective deployment in clinical and field operational settings. They summarize lessons learned in each area and look toward an optimal future. Although this work's overall interest and visibility, some aspects should still be considered to improve the quality and objectiveness.

In Conclusion, the authors should add the potential practical application.

Author Response

(The authors gave the same response as above.)

Reviewer 3 Report

Review of Manuscript “The evolution of medical countermeasures for Ebola virus disease: lessons learned and next steps“ by Ian Crozier et al..

This is a very well written and structured review of the medical countermeasures against Ebola virus disease (EVD), which have evolved especially in the 2013 to 2016 Western Africa outbreak and the 2018 to 2020 outbreak in the Democratic Republic of Congo. These countermeasures include improved diagnostic tools also applicable in near-treatment unit field labs, successful vaccination strategies and therapeutics. The recent advances in these areas are addressed in a very understandable and comprehensive manner and the conclusions drawn may also have important implications for the potential to improve responses to similar filovirus diseases such as those caused by Marburg virus or Sudan virus. I would therefore like to recommend publication, if the two minor points listed below are addressed in a revised version of the manuscript.

Minor points:

1) Numbering of the sections is not correct. Both Diagnostics and Vaccines including the corresponding sub-sections are listed as number 2.

2) In the section Therapeutics the authors should provide some more information on the pathogenesis of EVD including possible immune pathogenesis.

Author Response

(The authors gave the same response as above.)

Round 2

Reviewer 1 Report

thank you, still your manuscript needs an overall conclusion.

Author Response

Please see the attached PDF with our response.
